# Does your model understand genes? A benchmark of gene properties for biological and text models

## Abstract

The application of deep learning for biology, including foundation models, has increased significantly in recent years. Some models are text-based, while others are trained on the underlying biological data, especially omics data of various modalities. Consistently comparing the performance of deep learning models for biology has proven challenging due to the diversity of training data and downstream tasks. Here, we utilize the fact that many models operate on the level of genes and propose a unifying benchmark by defining hundreds of tasks based on ground-truth gene properties collected from professionally curated bioinformatics databases. We collect properties of five types: (1) genomic properties, including predicting which genes can be methylated or which are dose-dependent; (2) regulatory functions, evaluating how the genes participate in cellular regulatory processes; (3) localization, including identification of differential expression in different tissues or sub-cellular localization; (4) biological processes, including predicting gene involvement in pathways or disease prognostics; and (5) protein properties, including prediction of functional domains or post-translational modifications. These properties are used to define binary, multi-label and multi-class classification tasks. To create an architecture-agnostic benchmark we extract gene representation vectors from each model, including single-cell RNA-seq (scRNA) foundation models, large language models, protein language models, DNA foundation models, and classical baselines, and use them to train simple predictive models on the tasks. Depending on the model, we utilize the model's token-level embeddings of gene symbols or transform the gene symbol to an input appropriate for the model, i.e. a description of the gene for text models, the gene sequence for DNA models or amino acid sequences for the protein models. Using these embeddings on the benchmark tasks, we create a detailed assessment of the relative performance of the different models. In general, we find that text-based models and protein language models outperform the expression-based models on tasks related to genomic properties and regulatory functions, while expression-based models tend to outperform the others on localization tasks. We also observe performance for the classical bag-of-words baseline that is similar to the large language models for many tasks. By enabling broad systematic evaluation of diverse deep learning models in biology, this benchmark can help direct future research in artificial intelligence toward improved biological understanding and accelerated therapeutic discoveries. The code and benchmark data can be extended to more models and tasks and is available at GitHub.

## 1 Introduction

Recent successes in the application of self-supervised learning in natural language processing have given rise to foundation models, which are trained on a large unlabeled dataset and useful on a broad range of tasks (Bommasani et al., 2021). The potential to realize similar advances in biology has given rise to a new and rapidly growing cohort of biological foundation models, either as specialized language models or new models trained on biological modalities such as DNA sequences (Ji et al., 2021), amino acid sequences (Rao et al., 2021), electronic health records (Yang et al., 2022) or other

---

*These authors contributed equally to this work

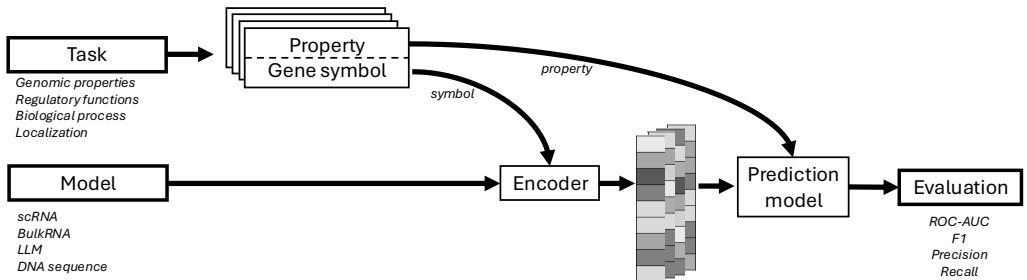

Figure 1: **Gene Benchmark evaluation flow** Diverse pretrained models are benchmarked based on the ability of their gene representations to predict gene properties as collected in tasks. For example, CREM is a transcription factor while KEAP1 is not. Depending on the model, gene representations may be token embeddings (as in transcriptomics) or they may be constructed from a gene sequence (textual, base-pair or amino acid sequence) that is encoded by the model. These vectors are then used to train a simple configurable predictive model for the task. The model performance after cross-validation represents the score for the pretrained model on the task. The tasks are described in Section 2 and code to run the pipeline shown above is available at GitHub.

modalities (Thieme et al., 2023). These models can identify functional sections of the genome, novel cell types, disease states, and more. Many of these efforts aim to identify the genes responsible for the various biological processes to identify future means to maintain these processes or restore their function. The proliferation of models and the potential for significant impact on human health gives rise to the need for robust evaluation and benchmarking. For text models, a number of biological/medical benchmarks have been published such as MedQA (Jin et al., 2021), and been widely adopted (Open Medical-LLM Leaderboard). Recent research provided a comparison of foundation models in specific modalities such as single cell models (Liu et al., 2023). However, no benchmarks have been proposed that compare foundation models across modalities or that compare text models against models trained on biological data directly. Part of the difficulty is that the downstream tasks that are used to compare scRNA FMs such as cell-type annotation, batch correction, and perturbation prediction (Ding et al., 2024) are very different from the benchmarking tasks used to evaluate NLP models, such as question-answering or sentence completion. The need for a benchmark that can work with text models and foundation models is particularly important in light of recent works such as Cell2Sentence (Levine et al., 2023), GenePT (Chen & Zou, 2023) and scInterpreter (Li et al., 2024), which have shown that text-models can be repurposed to work directly with transcriptomic data.

Here we propose to use *gene embeddings* to create a new benchmark that enables comparison of biological foundation models across modalities and against text models. Gene embeddings are an inherent component of expression-based foundation models built on Transformer architectures (Vaswani et al., 2017), parallel to word embeddings in text models. They can also be produced using the gene symbol or gene description with a language model supporting text embedding. Smaller models such as gene2vec (Du et al., 2019) or even bag-of-words models on textual descriptions of the gene can also produce gene embeddings. As with text embedding benchmarks, it is assumed that the models producing better gene embeddings are learning the ground truth more faithfully (Muennighoff et al., 2022).

To evaluate the gene embeddings, we compile a wide range of ground truth biological knowledge about genes including their genomic properties, regulatory functions, localization, their involvement in biological processes, and their protein properties (Table 1). We connect the gene embeddings to the relevant tasks, and evaluate their performance as illustrated in Figure 1. Though each task captures only a small part of the biology involving the gene, collectively they offer a multi-faceted, panoramic view of the gene. Superior performance on this collection of tasks thus implies that the model's learned embeddings are more inherently meaningful and thus useful for diverse downstream tasks even without seeing labeled data for these tasks.

We apply this benchmark to evaluate several families of models. These include text-based models, where the embeddings utilize large language models to encode a textual description of the genes, scRNA foundation models trained on multi-omics data, models that are based on protein or DNA sequence, and classical ML methods to act as a baseline comparison on text and gene expression data. Our analysis shows that text models outperform the other model families for most gene-related tasks, even when the information is not explicitly in the text. This result underscores the need and the potential to continue and improve knowledge integration into gene embeddings, thus improving gene target identification and all downstream tasks.

The benchmark platform is available (under an Apache 2.0 license) at GitHub, and includes scripts for task data downloads, as well as examples and documentation for using the benchmark on new models.

## 2 BENCHMARK TASKS

Following decades of work in bioinformatics and related fields, large amounts of structured data regarding genes have been compiled through projects such as Reactome (Milacic et al., 2023), Human Protein Atlas (Uhlen et al., 2010; Human Protein Atlas), OpenTargets (Ochoa et al., 2022) and HUGO Gene Nomenclature Committee at the University of Cambridge (Seal et al., 2022). This enables us to compile a wide variety of validated properties to use to test the quality of gene embeddings. Our benchmarking package allows defining the tasks in general terms, which allows simple addition of new tasks with multiple identifier types (see S6.1. Notably, the benchmarking package is not limited to gene-tasks and can be easily extended to other modalities.

### 2.1 TASKS DESCRIPTION

We compiled 312 gene properties, which we used to define evaluation tasks. Most of the tasks are based on single gene properties, while some are based on gene-pairs or links between genes and diseases.

For simplicity, we sort the properties into the following five families:

**Genomic properties** This family of tasks evaluates the ability to predict properties inherent to the gene sequence, including predicting which genes can be methylated, and which genes are dose-dependent (their expression depends on the number of copies in the genome) There is a total of 7 tasks in this family. See Table S1 for a full description.

**Regulatory functions** This family of tasks evaluates how the genes interact with other genes through the cellular regulatory processes and consists of a total of 6 tasks. These include predicting which genes are transcription factors, the number of connections in the gene-regulatory network, etc. See Table S3 for a full description.

**Localization** This family includes tasks for identifying differential expression and activity in different tissues or sub-cellular localization. That includes predicting protein levels found in blood, correctly assigning genes to expression clusters derived from various tissue samples, sub-cellular localization, etc. There are a total of 30 tasks in this family. See Table S4 for a full description.

**Biological processes** This family evaluates the biological functionality of the gene by evaluating tasks such as involvement in pathways, being prognostic of survival, and being associated with a disease. This family consists of 29 tasks, representing the most diverse set of questions. See Table S5 for a full description.

**protein properties** This family focuses on properties of the protein product of the gene, including its functional domains, post-translational modifications,and its ligands.

These properties cover many of the biological roles that genes play, providing an indication of how well a given pre-trained model has captured various aspects of gene representation, allowing for differentiation between various types of models and training data. Users can use the performance on different task families to select pre-trained models for their use-case.

Table 1: A breakdown of the number of tasks per prediction type and task family. Numbers in parenthesis represent the number of binary classification tasks that can be extracted from the multi-label tasks.

| Task family | Number of tasks (and sub-tasks) | | | | |
|---|---|---|---|---|---|
| | Binary | Multiclass | Multi-label | Regression | Total |
| Genomic properties | 3 | 1 | 3 (+79) | - | 7 (+79) |
| Regulatory functions | 5 | - | - | 1 | 6 |
| Localization | - | 21 | 1 (+70) | 7 | 30 (+70) |
| Biological processes | 3 | 21 | 3 (+91) | 2 | 29 (+91) |
| Protein properties | - | - | 3 (+53) | - | 3 (+53) |
| Total | 11 | 43 | 10 (+293) | 10 | 71 (+293) |

## 2.2 TASK ORIGIN

To benchmark the pre-trained models, we aimed to collect properties that are as diverse as possible, capturing the many roles that genes and the proteins they code play in biology. For reliability and reproducibility we have opted for gene properties that are manually validated by hand and freely available, see Section S6.1.6 for data availability details.

**Reactome** The pathways tasks were curated by taking the full list of genes from the Human Genome Nomenclature Committee (HGNC) downloadable files (including protein-coding gene, non-coding RNA, pseudogene and other tables) (Seal et al., 2022) and labeling each symbol by its inclusion in a top-level pathway S6 from Reactome (Milacic et al., 2023).

**Human Protein Atlas** Protein atlas (Human Protein Atlas) tasks were created by compiling the protein atlas file v23 (Protein Atlas Data v23). We selected columns that contained features regarding gene properties, sorting them to binary, multiclass, multi-label, or regression tasks. We removed rows with missing data or genes that had no symbol name.

**Open Targets** The gene-disease association (Ochoa et al., 2022) task was compiled by downloading the overall association score of genes and diseases from the open targets platform (we used the direct file interface using the files published on 2023-09-21 version 23).

**Uniprot** Three protein properties tasks were created using data from the Universal Protein Knowledgebase (UniProt) by assigning binary labels to gene symbols if a given keyword value is present for any of the protein products of that gene symbol. The keyword categories used were Domain, Ligand and Post-transcriptional modification (Consortium, 2022).

**Publications** Since several papers have used gene properties to evaluate pretrained models, we included those tasks in our benchmark (Chen & Zou, 2023; Fang et al., 2024; Lambert et al., 2018). There are 9 tasks derived directly from publications.

These tasks cover a wide variety of gene roles and provide a well-grounded assessment of gene representation quality. Because research in this domain evolves quickly, we have provided the ability to extend our benchmark with more tasks, at GitHub.

## 2.3 TASK DEFINITION

Independent of the task family, each task evaluates a specific outcome type: a) a binary , or b) a multi-label assignment, or c) a multiclass, or d) a regression task. We used gene properties to define tasks only if at least 1% of the covered entities had the label. Binary sub-tasks were derived from multi-label tasks by selecting specific labels. For all tasks, we used the gene symbol as an identifier; ensemble stable IDs were converted into symbols using MyGeneInfo (Wu et al., 2012). To simplify comparisons between models we limited the scope of each task to the gene symbols shared by all encoding models.

Table 2: Description of the prediction models, evaluation metrics, and cross validation scheme used for each of the four task types

| Task Type | Prediction Model | Metric | Cross-validation |
|---|---|---|---|
| Binary | Logistic regression | AUC-ROC, F1, Precision, Recall, Accuracy | Stratified cross-validation |
| Multiclass | Logistic regression | AUC-ROC one versus rest, F1, Precision, Recall, Accuracy | Stratified cross-validation |
| Multi-label | Multiple output logistic regression | AUC-ROC, Hamming, F1, Precision, Recall, Accuracy | K-fold |
| Regression | Linear regression | R-squared, RMSE, mean absolute error | K-fold |

## 2.4 TASK EVALUATION

In contrast to text embedding benchmarks such as MTEB (Muennighoff et al., 2022) where the quality of the model is assessed by its ability to generate similar embeddings for known similar texts, evaluating the biological properties of genes embeddings requires a slightly different approach. Because the genes have many biological properties, we cannot assess the model quality by similarity alone. For this reason, we have primarily adopted classification metrics: the embeddings are provided as inputs to a simple logistic or linear regression model to predict the ground truth properties, and evaluated with 5-fold cross-validation. The benchmark can also be defined using non-linear models, which could detect information in the representation vectors more successfully than a linear model, we discuss this and assess the differences in Section 5.

This setup enables us to evaluate whether the correct information is encoded in the vector without making a-priori assumptions about the underlying information properties of the embedding space.

## 3 ENCODING MODELS

We selected several publicly available models for comparison from five major families: Large language models trained on text, deep-learning models trained on gene expression data, deep learning models trained on base pair sequences of genes, deep learning models train on amino acid sequences and classical machine-learning models. We used models that were openly available with weights. When available, we used top-performing models according to independent leaderboards. Table 3 provides a summary table of model properties, and the following is a brief description of each model. The gene-benchmark allows for simple integration of additional models and tasks (see Supplementary text S6.1)

### 3.1 TEXT BASED MODELS

For text embedding models, we create an embedding for a gene by extracting the standard symbol, full name, and description of the gene from the *NCBI Entrez Gene database* (Maglott et al., 2010). This information is packed into a textual description that is given to the model as a prompt in the format `"Gene symbol <symbol> full name <full name> with the summary <summary description>"`, and this prompt is embedded using a sentence embedding model. As a result, the benchmark that we have defined works seamlessly with any model supported by `sentence_transformers` (Reimers & Gurevych, 2019). For this assessment, we selected the top performing models from the leading embedding benchmark, the MTEB leaderboard (MTEB Leaderboard) and from the sentence transformers leaderboard (Sentence Transformers Leaderboard). For simplicity and ease of replication, we limited ourselves to models that did not require to trust remote code as defined by the `sentence_transformers`

API (`trust_remote_code` set to false). In addition to compare performance with a simpler non-parametric method, we used a Bag-of-words encoder

**MTEB-L** A variant of *Mistral 7B* (Meng et al., 2024) called *SFR-Embedding-Mistral*, which is a transformer based generative LLM with 7.11B parameters. Chosen as the top performing open model on MTEB (MTEB Leaderboard) as of May 2024.

**MTEB-S** A compact sentence-embedding model with 335M Parameters (Lee et al., 2024) called *mxbai-embed-large-v1*. Chosen as the top performing small open model (<1B parameters) on MTEB (MTEB Leaderboard) as of May 2024.

**MPNet** A transformer-based textual LLM (Song et al., 2020), pre-trained on over 160GB text corpora. Pretraining was done using masked and permuted language modeling learning. It was chosen since it was the top performing model on sentence transformers (Sentence Transformers Leaderboard) as of May 2024.

**Bag-of-words** A statistical word-based text model which does not take into account the order of words in the text. The presence of each word is used as an independent feature. We used `CountVectorizer` from scikit-learn (Pedregosa et al., 2011), with default parameters to select the top informative 1024 words, and used the word counts vector as the embedding for each description. The model was fitted to the text of the gene descriptions.

### 3.2 GENE EXPRESSION AND TRANSFORMER-BASED MODELS

Inspired by the success of transformer-based LLMs in NLP, these models aim to learn biology as a "language" over scRNA-seq readings, fitting an embedding to each gene as if it were a 'word' in an NLP model, and the transformer based architectures integrate the gene expressions into cell-level embeddings. We made use of a recent survey and benchmark to highlight the three best performing open scRNA foundation models (Liu et al., 2023). The gene embeddings were extracted from the publicly available model weights. Gene names were taken from the supplementary model configuration files.

**CellPLM** A transformer-based foundation model for single-cell biology with over 80M parameters (Wen et al., 2023). Trained on scRNA-seq and spatially resolved transcriptomic (SRT), adding tissue level information. Trained using MLM variant, on 9 million scRNA-seq cells and 2 million SRT cells. Embedding extracted from the `embedder.feat_enc.emb` layer in the model downloaded (Wen et al.), with the gene names from the matching configuration file.

**Geneformer** A transformer based foundation model for single cell biology with 10.3M parameters. (Theodoris et al., 2023a). This model represents the scRNA expression using a list of genes ranked by their normalized expression levels. This is intended to make the order significant, and allows the use of context-aware attention mechanisms similar to these that work well in NLP. The model is trained on about 30M scRNA-seq readings. Embedding extracted from the `embeddings.word_embeddings` layer from (Theodoris et al., 2023b)

**ScGPT** A generative foundation model for single-cell transcriptomics utilizing a self-attention, with 53M parameters (Cui et al., 2024). Pretrained using masked language model (MLM) training. Explicitly encoded genes, expression levels and conditions, concatenated to represent each gene in context. Training is performed using a masked language modeling variant, where masking is done with attention masking to accommodate for the non-sequential nature of the data. Embedding extracted from (Cui et al., b) following the instructions in (Cui et al., a), steps 1 and 2. We used two variants, blood (designated ScGPT-B) trained on 10.3 million blood and bone marrow cells and the human model (designated ScGPT-H) trained on 33 million normal human cells.

**Gene2vec** A 200 dimensional concept embedding of the human genes (Du et al., 2019), based on the concept of Word2Vec (Mikolov et al., 2013) and learned from co-expression patterns, shared Gene Ontology (GO) annotation, tissue-specific genes, and functional gene sets.

Table 3: Summary descriptions of the gene-encoding models

| Model | Input type | Model type | Num of params | Output size |
|---|---|---|---|---|
| MTEB-L | Text | Transformer | 7.1B | 4,096 |
| MTEB-S | Text | Transformer | 109M | 1024 |
| MPNET | Text | Transformer | 420 M | 768 |
| Bag-of-words | Text | Non-parametric | - | 1,024 |
| CellPLM | ScRNA-seq | Transformer | 85M | 1024 |
| Geneformer | ScRNA-seq | Transformer | 10.3M | 256 |
| ScGPT-H | ScRNA-seq | Transformer | 51M | 512 |
| ScGPT-B | ScRNA-seq | Transformer | 39M | 512 |
| Gene2Vec | Bulk RNA-seq | Word2Vec | 5M | 200 |
| DNABERT-2 | Base pair sequence | Transformer | 117M | 768 |
| ESM-2 | Protein sequence | Transformer | 3B | 2560 |

## 3.3 BASE-PAIR MODELS

Every gene can be mapped to its DNA sequence. There have been numerous recent advances in foundation models trained on DNA. Though not trained specifically to represent genes, by representing DNA they can generate gene representations as well. **DNABERT-2** A BERT based genome foundation model (Zhou et al., 2024) trying to decode a linguistic representation of the genome. In this method they replaced the common k-mer tokenization with a Byte Pair Encoding (BPE) tokenization. This model performed well in a recent DNA model benchmark (Liu et al., 2024).

## 3.4 PROTEIN LANGUAGE MODELS

Representing the protein products of a gene, may be considered a representation of the gene itself. Following the approach of SATURN (Rosen et al., 2024) and UCE (Rosen et al., 2023), we have represented the gene symbol as the mean of its protein product representation vectors.

**Evolutionary Scale Modeling-2 (ESM-2)** SOTA general-purpose protein language model. A transformer model trained on sequences of natural proteins (Lin et al., 2023) which is able to generate novel proteins. The model was trained using the ESM Metagenomic Atlas that contains >617 million metagenomic protein sequences. We took the model `esm2_t36_3B_UR50D` with 3 billion parameters.

## 4 RESULTS

We report our gene-benchmarks on eleven models, evaluated on all tasks. The benchmark results demonstrate that the various models exhibit different performance patterns on different tasks. When grouping the performance measures by family tasks and averaging across tasks we find that the four text-based models exhibit better performance for the genomic properties and regulatory function families, while the scRNA-based models performed better at the localization and biological process tasks (Figure 2). These trends are consistent when using other evaluation metrics such as F1 (see Figure S1). The calculation time for the benchmark varies depending on the model size. For the largest model, with 7B parameters and embedding size 4096 (Table 3), creating the gene embeddings took approximately 40 minutes on a single NVIDIA A100 80GB. The calculation time for fitting the predictive models is highly dependent on the embedding size, with the smaller embeddings requiring less than an hour to calculate all the benchmarks and the largest requiring 15+ hours on a 48-core Xeon E5.

Interestingly, text-based models using transformer architecture only slightly outperform the bag-of-words model in most tasks. Furthermore, we do not see an advantage to the model size, where the MTEB-L model exhibits comparable performance to the smaller MTEB-S and MPNet models. Similarly, in the scRNA-based models the transformers usually slightly outperform the older gene2vec model, which is based on word2vec architecture and trained on bulk RNA expression data. ScGPT-H

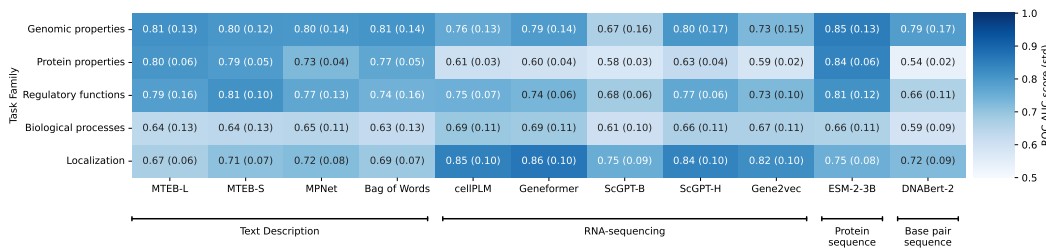

Figure 2: The performance of each model on the task families as measured by average area under the ROC curve. Parentheses show the corresponding standard deviation across all tasks of the same family.

was the top performer in two different families of tasks. Here, too, we do not see a clear advantage for larger models, with cellPLM exhibiting comparable performance to the smaller ScGPT-H and even smaller Geneformer. ScGPT-H outperformed ScGPT-B significantly, which is likely due to the larger, more diverse, training data. Indeed, it is to be expected that a model trained on expression data from a single tissue would not perform well on tissue localization tasks related to other tissues.

A closer examination of the mean AUC per task (Figure S2, Figures S3, S4, S5, S6, S7, S8, S9) reveals a more complex picture, where within each task family some tasks are dominated by text-based models and others by expression-based models or protein language models. This can also be seen by the high cosine similarity observed between overall task performance amongst models from the same type, as exhibited in Figure S10.

The protein models perform best at protein properties, but less so for biological processes and localization. DNABert-2 performs worse than the other models for all but the genomic properties, where it is comparable to the other model families.

The tasks themselves also show a clustering in performance, as shown in Figure S11, but also show a large range of dissimilarity, suggesting that the benchmark tasks correspond to distinct biological phenomena.

Above we used linear models for predicting gene properties from vectors. We consider the possibility that a more expressive model could perform better by training a multilayer perceptron model (MLP) on the binary tasks, comparing the MTEB-L and MTEB-S embeddings. We find that the performance is closely correlated to logistic regression, as seen in Figure S12. For this reason, we prefer the linear models which are not sensitive to hyperparameter selections and thus enable robust comparisons across many thousands of combinations of models and tasks. Nevertheless, the benchmark package code at GitHubsupports the use of any scikit-learn model.

One notable result is that text models outperform the scRNA models in most disease involvement tasks except in the Pathology tasks, chromosome, and N1 Network, indicating that there are exceptions to the general rules of model performance we outlined. Similarly, expression-based models outperform in cell-type localization tasks, but under-perform in sub-cellular localization tasks. This is in line with our expectations, given the close relation between cell-type, tissue-type and single cell RNA expression levels.

## 5 SUMMARY AND DISCUSSION

We present a gene-centric benchmark that includes hundreds of tasks, sorted into functional families. We designed this benchmark to evaluate the gene embeddings provided by pretrained models applied to biology, thus suggesting a common ground for evaluating the potential of the models to provide useful, and potentially novel insights on gene involvement in biological and medical questions.

We applied the benchmark to a representative set of models trained on text, gene-expression data, protein sequences and DNA sequences and compared their performance. We observed that each family of models exhibits superiority for a different set of tasks, hinting that combining the knowledge that comes from multiple modalities may provide additional benefits. It should be pointed out,

however, that our analysis is not intended to be a comprehensive comparison of all text models and all gene-expression models and that our results may not be generalizable to all such models.

The embeddings that we chose to use for the text models were based on a detailed description of the genes, rather than using the gene symbol, which could have evaluated whether the LLM has sufficient useful information on genes from its general training on text. However, while virtually all coding genes have literature in PubMed, the coverage is highly skewed, with the most popular 10% of genes accounting for over 60% of the publication (Lee et al., 2019), suggesting that language model pretraining may be skewed towards the more widely studied genes. Instead, in our evaluation of text models, we included a bag-of-words model that can serve as a baseline of sorts, allowing us to examine the ability of LLMs to generalize and contextualize. We find that in most tasks the LLMs provide a modest improvement in performance, indicating that additional work may be needed to create LLMs that are useful for basic biology.

We point out that most of the models we evaluated here were not designed to provide useful gene embeddings. Some were trained to perform textual tasks, while others were trained to predict properties at the whole-cell level. However, utilizing pretrained gene embeddings from a foundation model to improve performance of a more specialized model has become an active research area with numerous promising results produced in the last year. For example, scFoundation (Hao et al., 2023) showed improvement on gene perturbation prediction by injecting their gene embeddings into GEARS (Roohani et al., 2023), SATURN (Rosen et al., 2024) has used ESM (Lin et al., 2023) protein embeddings to enable cross-species cell label propagation and GNN based models such as Otter-Knowledge (Lam et al., 2023) and BioBridge (Wang et al., 2023) have proposed comprehensive biomedical models built on embeddings across multiple domains. Given the interest and promise of this technology, our benchmark can help guide researchers toward more successful application of deep learning to biology.

## 5.1 Limitations and Future Work

We gathered the benchmark tasks from actively maintained professionally curated sources and we have relied on their quality control processes. For many reasons, the entire genome is not studied evenly (Lee et al., 2019) and when genes are studied, the full diversity of human ancestry is not evenly reflected (Fairley et al., 2019). As biological research improves in performance and fairness, we look forward to updating our benchmark tasks accordingly. We used only open source models with released weights excluding models that did not (Zrimec et al., 2022). Though we have explored the benchmark tasks using gene embeddings, the tasks could be utilized in other ways, such as by defining fine-tuning objectives for deep learning models or even as the basis for question answering in text models. Such a strategy, while not applicable for all models, may uncover predictive power that is specific to each model.

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

## 6 Appendix

### S6.1 The Gene-Benchmark Package

This freely available package was developed to facilitate easy access to the tasks and efficient use of them for benchmarking. It includes three main modules as well as notebooks and scripts that demonstrate the package's usability. The main flow of task evaluation using the package is described in Figure 1. Below we review the main modules used in the package, which is available at GitHub.

#### S6.1.1 Tasks

This module contains two main parts. The first is the means to load the task definition according to task name in a generic format into a designated, easy-to-use object. The class allows easy access to the entity identifiers (usually gene symbols) and their outcomes. They will usually be a single columned data frame, but if the task includes multiple genes per instance (for example, gene-to-gene interaction), it will include multiple entities in a column structure. In the multi-label case, the output is also a multi-columned data frame. The second part is a pipeline class that manages the process from a task name to description (in the case of text-based models) to encoding, training a simple prediction model in a cross-validation fashion, and creating a report. Adding additional tasks is designed to be simple, all it requires is saving the task descriptions in a specific format.

#### S6.1.2 Descriptor

The module manages the transition from an entity identifier into a text description. For gene symbols, we retrieve the description fields from NCBI using the *MyGene.Info* services and construct a description sentence. We allow predefined descriptions by creating a descriptor that loads the descriptions from a CSV file. This feature enabled us to download the disease description from open targets without needing to integrate with their service and facilitate easy introduction of new descriptions. We are also able to construct multi identifier types descriptors, thus enabling the creation of a descriptor that can describe tasks with different identifiers, such as in gene-disease association.

#### S6.1.3 Encoder

This module manages the encoding of either the entity identifier or its textual summary. We enable encoding using any HuggingFace sentence transformer supporting module. In addition, we enable encoding using a pre-computed encoder by loading the encodings from a precomputed CSV file. This enables us to pre-compute the encoding from scRNA-based models. In addition, we enable the creation of a multi-entity type encoder that enables encoding each type of entity differently. For example, in the case of Gene-Disease association, we can encode the genes using pre-computed encoding and the disease using a sentence transform encoder.

#### S6.1.4 Base models

The package supports any scikit-learn model. For the manuscript, we explored linear and logistic regression with the default scikit-learn parameters, and an MLP with three hidden layers of size 100 and 500 max iterations.

#### S6.1.5 Scripts and Notebooks

To efficiently create benchmarks the package includes a command line interface. Enabling benchmarking multiple models (described in YAML format) on multiple tasks (supplied in the command line or in YAML) and output a single report in CSV format. An additional script is supplied that can extract the embedding of the given identifiers list. The package also includes a notebook demonstrating how the package can be used and how to create figures, as displayed in this manuscript.

#### S6.1.6 Data availability and licensing

All of the data is from publicly available sources and the steps required to download and prepare the tasks for benchmarking are implemented in our GitHub repository. We did not produce the task data

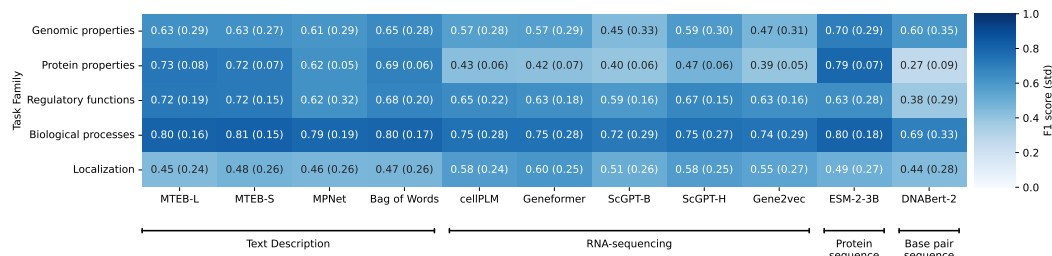

Figure S1: The performance of each model on the task families as measured by average f1 score. Parentheses show the corresponding standard deviation across all tasks of the same family.

and do not redistribute the data used for the benchmark tasks. To reproduce the results shown here, we provide code to populate the benchmark task directly from the public sources. We do not own the task data and refer the users to the licenses of the data owners.

- **Reactome** - The current task retrieval code downloads that pathway directly from reactome's current server, Reactome data is robustly backed at third party servers. Reactome content is readily accessible for download from its website, GitHub, and various aggregators like NCBI and EMBL-EBI with data availability path in case of of funding loss. see Reactome digital preservation for further details.

- **The human protein atlas** - Tasks are created using files saved at the human protein atlas, the data is accessible using programmatic access as well. Previous versions of the data files are accessible as well.

- **Open Targets** - open targets is committed to open source and supporting open access research. with multiple data download and retrieval options see data access for further details provides data from community contributions, with the original data owners retaining ownership and rights. There are no additional restrictions on the use or redistribution of this data, Open Targets allow does not guarantee the accuracy or suitability of the data or services provided

- **Uniprot tasks** UniProt conforms with EMBL-European Bioinformatics Institute's data preservation policies. Uniprot has applied a CC-BY-4.0 license to the copyrightable parts of their database. For more info see Uniprot license.

- **Publication tasks** - The data used for the creation of these tasks comes from the cited publications. For the HLA task we the data was derived from the HGNC web site. For the Tf vs non-tf task the data was derived from the The Human Transcription Factors web-site. That made the data publicly available via files but did not make clear data availability commitment. Scripts detailing exactly how to obtain these datasets and to construct the tasks as utilized here are provided at GitHub.

Table S1: Detailed description of the genomic tasks used for benchmarking

| Task Name | Type | Size | Description | Origin |
|---|---|---|---|---|
| Bivalent vs non-methylated | binary | 133 | Does the gene go through methylation or is it bivalant | Geneformer |
| Chromosome | categorical | 19784 | Chromosome | human protein atlas |
| Dosage sensitive vs insensitive tf | binary | 487 | Is gene expression affected by the number of copies it has | Geneformer |
| Lys4-only-methylated vs non-methylated | binary | 171 | Does gene go through Lys4 methylation | Geneformer |
| Protein class | multi label | 19784 | Protein class(es) of the gene product according to selected gene lists | human protein atlas |

Table S2: Detailed description of the protein structural tasks used for benchmarking

| Task Name | Type | Size | Description | Origin |
|---|---|---|---|---|
| UniProt Keyword PTM | multilabel | 13968 | Predicts which chemical modifications occur after protein synthesis (e.g., phosphorylation, glycosylation). | UniProt |
| UniProt Keyword Ligand | multilabel | 6673 | Predicts specific molecules or ions the protein can bind to, impacting its function (e.g., ATP, zinc). | UniProt |
| UniProt Keyword Domain | multilabel | 13469 | Predicts structural or functional regions within the protein, like specific repeats or conserved motifs (e.g., transmembrane domain). | UniProt |

Table S3: Detailed description of the regulatory tasks used for benchmarking

| Task Name | Type | Size | Description | Origin |
|---|---|---|---|---|
| Gene2gene | binary | 290032 | Pairs of genes with a known association | GenePT |
| Interactions | regression | 11348 | Number of proteins each gene is known to interact with | human protein atlas |
| Long vs short range tf | binary | 174 | Is activation by the TF binary (short range) or linear (long range) | GenePT |
| N1 network | binary | 1103 | Division of the N1 gene regulatory network into core and peripheral downstream effectors | How do Large Language Models understand Genes and Cells |
| N1 targets | binary | 281 | Genes that are downstream targets in the N1 gene regulatory network | How do Large Language Models understand Genes and Cells |
| Tf vs non-tf | binary | 2765 | Is the gene known to function as a transcription factor | The Human Transcription Factors |

Table S4: Detailed description of the localization tasks used for benchmarking

| Task Name | Type | Size | Description | Origin |
|---|---|---|---|---|
| Blood concentration - conc. blood im | regression | 438 | Concentration of protein in blood stream | human protein atlas |
| Blood expression cluster (HPA) | categorical | 12697 | Cluster assignment in blood-derived expression data | human protein atlas |
| Brain expression cluster (HPA) | categorical | 17590 | Cluster assignment in brain-derived expression data | human protein atlas |
| Cell line expression cluster (HPA) | categorical | 19167 | Cluster assignment in cell-line expression data | human protein atlas |
| RNA blood cell distribution | categorical | 19784 | Classification of spatial distribution of gene by histochemistry | human protein atlas |
| RNA blood cell specificity | categorical | 19784 | Level of differential expression compared to other tissues | human protein atlas |
| RNA blood cell specificity score | regression | 2238 | Fold change between highest expression the second highest expression | human protein atlas |
| RNA blood lineage distribution | categorical | 19784 | Classification of spatial distribution of gene by histochemistry | human protein atlas |
| RNA blood lineage specificity | categorical | 19784 | Level of differential expression compared to other tissues | human protein atlas |
| RNA blood lineage specificity score | regression | 3351 | Fold change between highest expression the second highest expression | human protein atlas |
| RNA brain regional distribution | categorical | 19784 | Classification of spatial distribution of gene by histochemistry | human protein atlas |
| RNA brain regional specificity | categorical | 19784 | Level of differential expression compared to other tissues | human protein atlas |
| RNA brain regional specificity score | regression | 595 | Fold change between highest expression the second highest expression | human protein atlas |
| RNA cell line distribution | categorical | 19784 | Classification of spatial distribution of gene by histochemistry | human protein atlas |
| RNA cell line specificity | categorical | 19784 | Level of differential expression compared to other tissues | human protein atlas |
| RNA cell line specificity score | regression | 2171 | Fold change between highest expression the second highest expression | human protein atlas |
| RNA mouse brain regional distribution | categorical | 16655 | Classification of spatial distribution of gene by histochemistry | human protein atlas |
| RNA mouse brain regional specificity | categorical | 16655 | Level of differential expression compared to other tissues | human protein atlas |
| RNA pig brain regional distribution | categorical | 16595 | Classification of spatial distribution of gene by histochemistry | human protein atlas |
| RNA pig brain regional specificity | categorical | 16595 | Level of differential expression compared to other tissues | human protein atlas |
| RNA single cell type distribution | categorical | 19761 | Classification of spatial distribution of gene by histochemistry | human protein atlas |
| RNA single cell type specificity | categorical | 19761 | Level of differential expression compared to other tissues | human protein atlas |
| RNA single cell type specificity score | regression | 4798 | Fold change between highest expression the second highest expression | human protein atlas |
| RNA tissue cell type enrichment | multi label | 13957 | The tissues in which the genes is significantly expressed | human protein atlas |
| RNA tissue distribution | categorical | 19784 | Classification of spatial distribution of gene by histochemistry | human protein atlas |
| RNA tissue specificity | categorical | 19784 | Level of differential expression compared to other tissues | human protein atlas |
| RNA tissue specificity score | regression | 4653 | Fold change between highest expression the second highest expression | human protein atlas |
| Single cell expression cluster | categorical | 19016 | Assignment to a cluster according to single-cell expression profile across tissues | human protein atlas |
| Subcellular location | multi label | 13039 | Subcellular Localization according to immunocytochemistry/IF | human protein atlas |
| Tissue expression cluster | categorical | 18355 | Assignment to a cluster according to bulk RNA expression profile across tissues | human protein atlas |

Table S5: Detailed description of the biological tasks used for benchmarking

| Task Name | Type | Size | Description | Origin |
|---|---|---|---|---|
| Biological process | multi label | 10796 | UniProt keywords indicating involvement in a particular biological process | human protein atlas |
| Ccd protein | binary | 1429 | Cell cycle dependent (CCD) proteins in the FUCCI U-2 OS cell line | human protein atlas |
| Ccd transcript | binary | 1631 | Cell cycle dependent (CCD) genes by RNA expression in the FUCCI u-2 OS cell line | human protein atlas |
| Disease involvement | multi label | 5837 | UniProt keywords for disease, cancer, and FDA approved drug targets | human protein atlas |
| Gene-disease association | regression | 411569 | Disease-gene association score as derived from the open targets platform | open targets |
| Hla class i vs class ii | binary | 44 | Identify class1 or class2 of the Histocompatibility complex (HLA) | ScGPT/HGNC |
| Molecular function | multi label | 10991 | Keywords assigned by UniProt to proteins due to their particular molecular function. | human protein atlas |
| Pathology prognostics | categorical | 0 | Evidence level for being prognostic of survival in 17 cancer types (17 tasks) | human protein atlas |
| Pathways | multi label | 10969 | Reactome top level pathways that include the gene (29 pathways) | reactome |
| RNA cancer distribution | categorical | 19588 | Classification of spatial distribution of gene by histochemistry | human protein atlas |
| RNA cancer specificity | categorical | 19588 | Level of differential expression compared to other tissues | human protein atlas |
| RNA cancer specificity score | regression | 2084 | Fold change between highest expression the second highest expression | human protein atlas |
| Secretome function | categorical | 2736 | Functional annotation of involvement in cellular secretion | human protein atlas |
| Secretome location | categorical | 2767 | The site of cellular secretion (if any) | human protein atlas |

Table S6: The list of top level reactome pathway names and unique identifiers

| Pathway name | Pathway identifier |
| --- | --- |
| Autophagy | R-HSA-9612973 |
| Cell Cycle | R-HSA-1640170 |
| Cell-Cell communication | R-HSA-1500931 |
| Cellular responses to stimuli | R-HSA-8953897 |
| Chromatin organization | R-HSA-4839726 |
| Circadian Clock | R-HSA-400253 |
| Developmental Biology | R-HSA-1266738 |
| Digestion and absorption | R-HSA-8963743 |
| Disease | R-HSA-1643685 |
| DNA Repair | R-HSA-73894 |
| DNA Replication | R-HSA-69306 |
| Drug ADME | R-HSA-9748784 |
| Extracellular matrix organization | R-HSA-1474244 |
| Gene expression (Transcription) | R-HSA-74160 |
| Hemostasis | R-HSA-109582 |
| Immune System | R-HSA-168256 |
| Metabolism | R-HSA-1430728 |
| Metabolism of proteins | R-HSA-392499 |
| Metabolism of RNA | R-HSA-8953854 |
| Muscle contraction | R-HSA-397014 |
| Neuronal System | R-HSA-112316 |
| Organelle biogenesis and maintenance | R-HSA-1852241 |
| Programmed Cell Death | R-HSA-5357801 |
| Protein localization | R-HSA-9609507 |
| Reproduction | R-HSA-1474165 |
| Sensory Perception | R-HSA-9709957 |
| Signal Transduction | R-HSA-162582 |
| Transport of small molecules | R-HSA-382551 |
| Vesicle-mediated transport | R-HSA-5653656 |

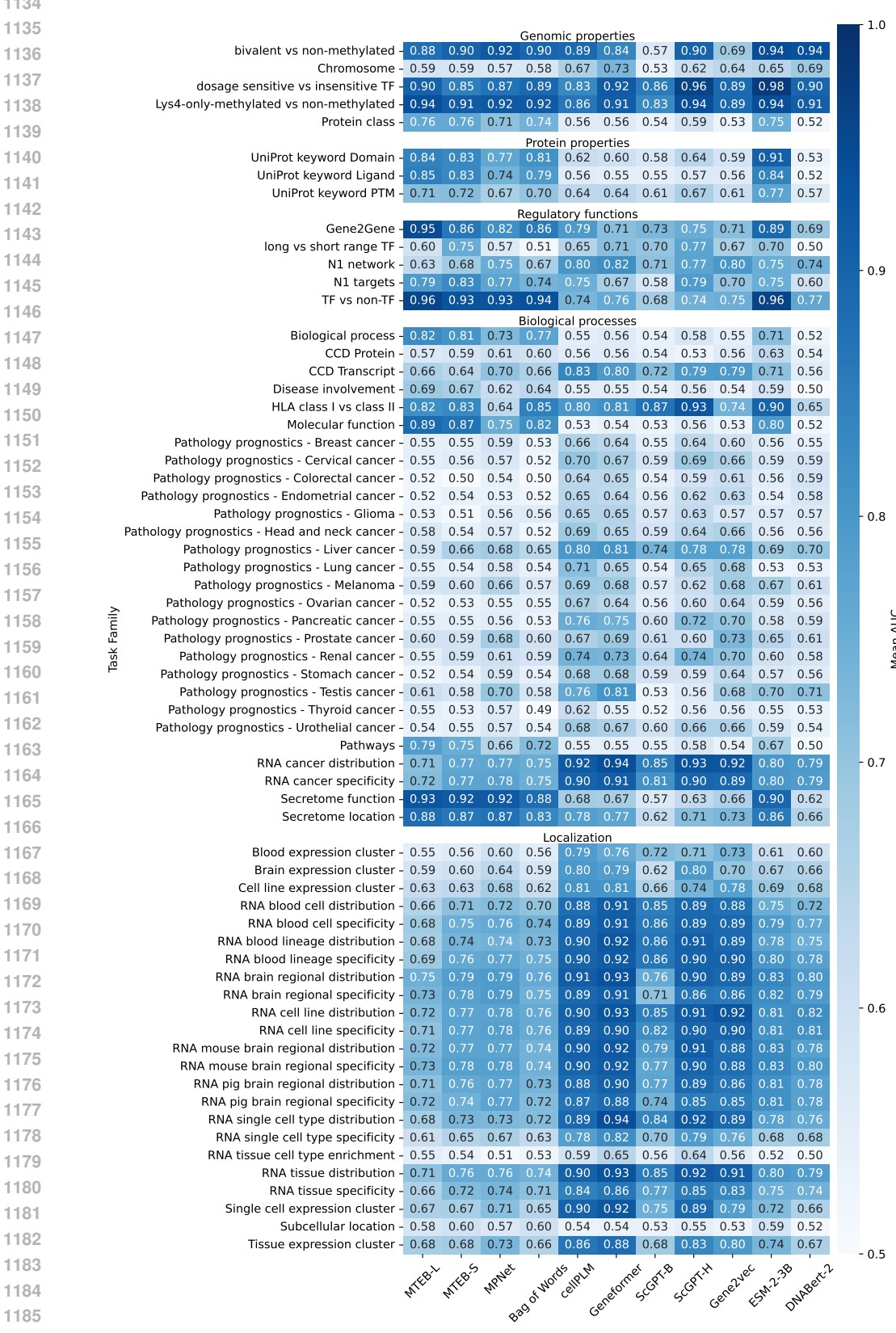

Figure S2: Mean AUC per model and task

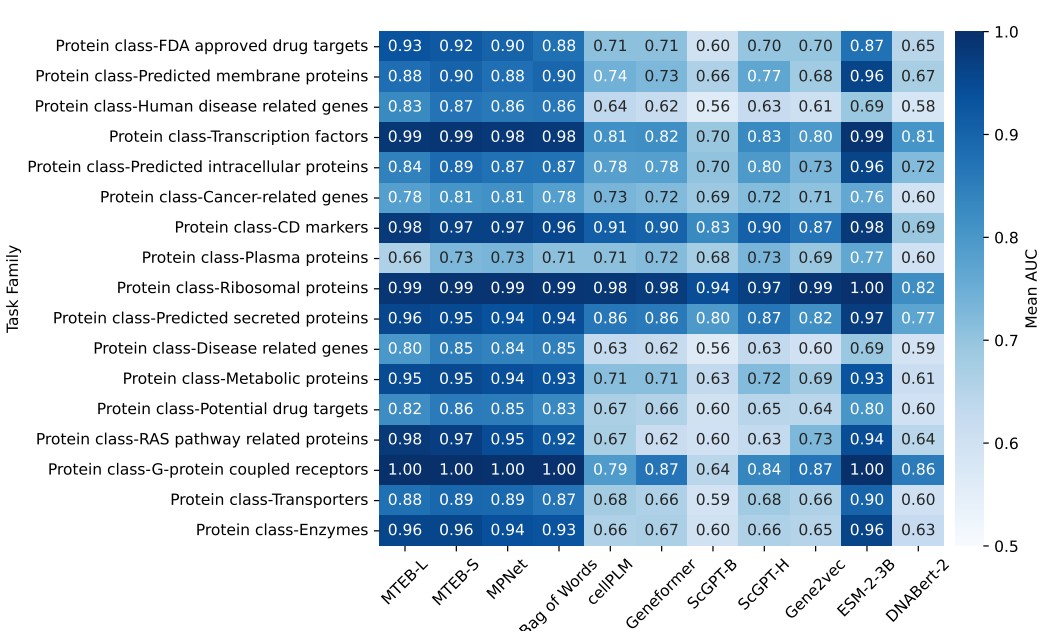

Figure S3: Model performance measured by mean AUC for binary tasks derived from the multi label task 'protein class'

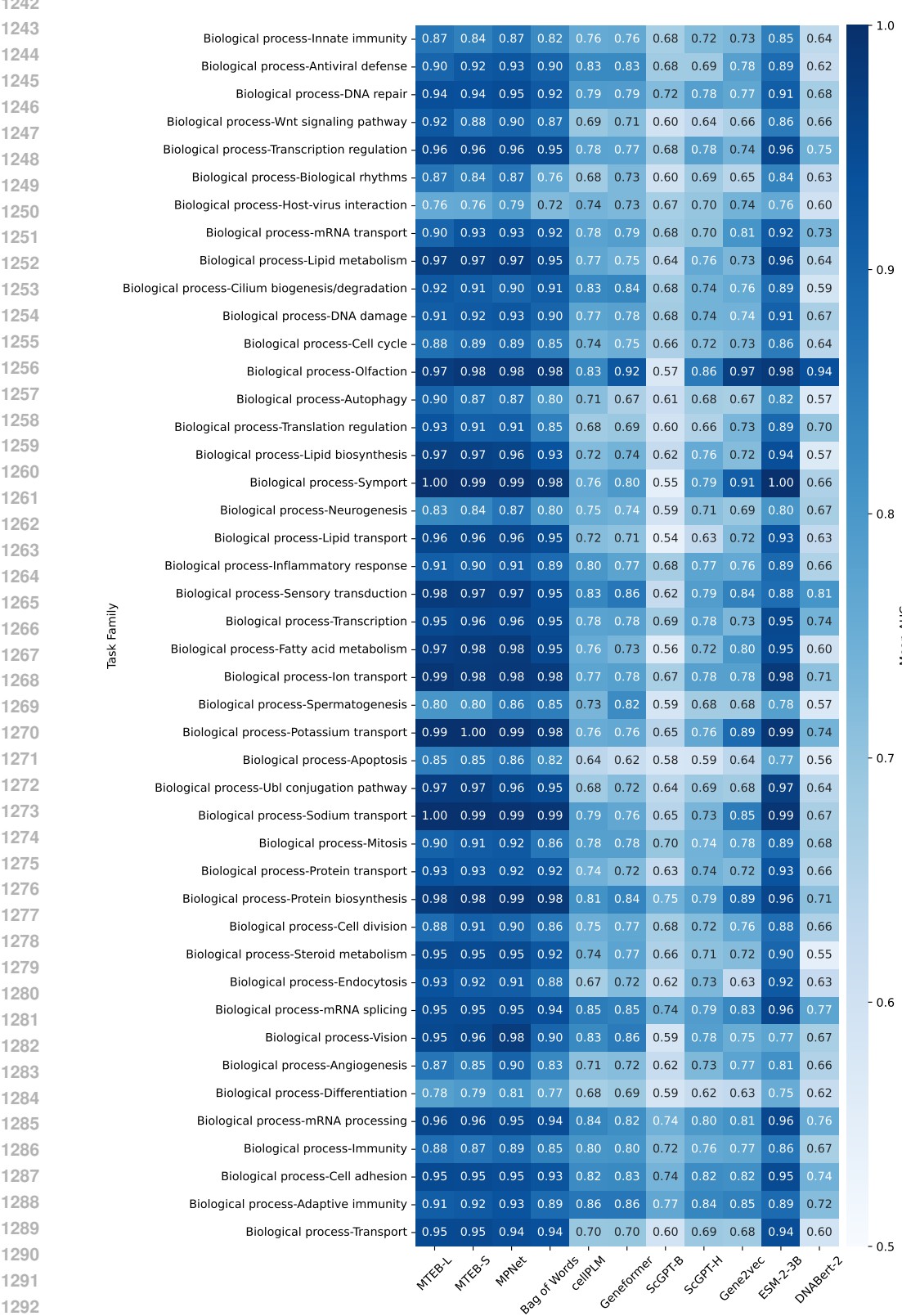

Figure S4: Model performance measured by mean AUC for binary tasks derived from the multi label task 'biological process'

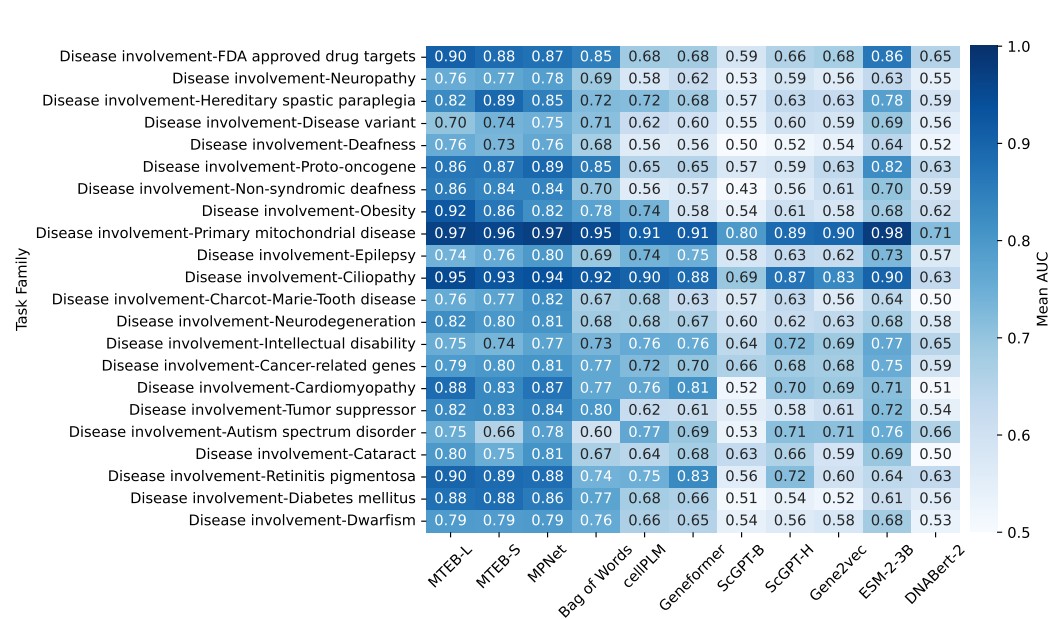

Figure S5: Model performance measured by mean AUC for binary tasks derived from the multi label task 'disease involvement'

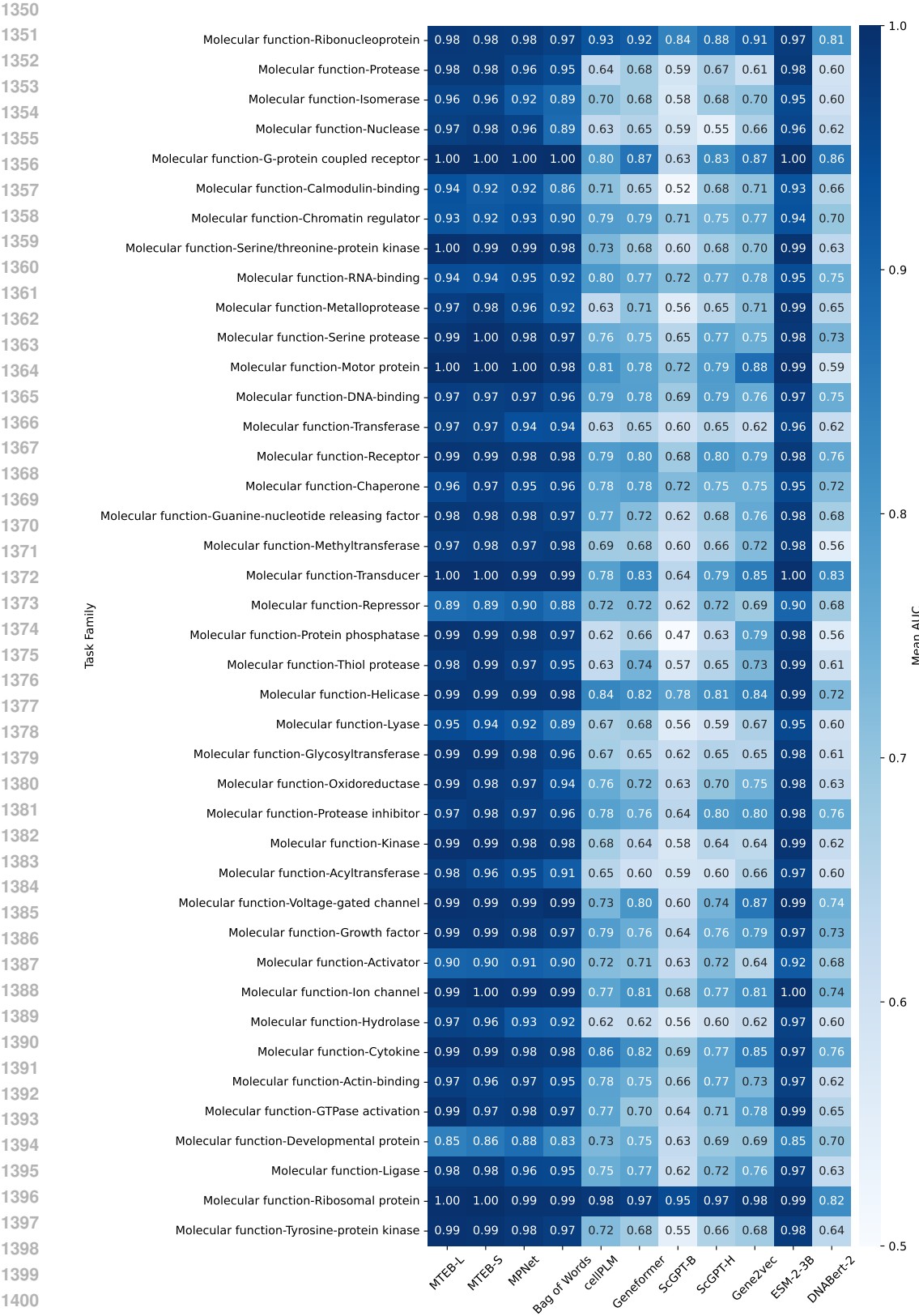

Figure S6: Model performance measured by mean AUC for binary tasks derived from the multi label task 'molecular location'

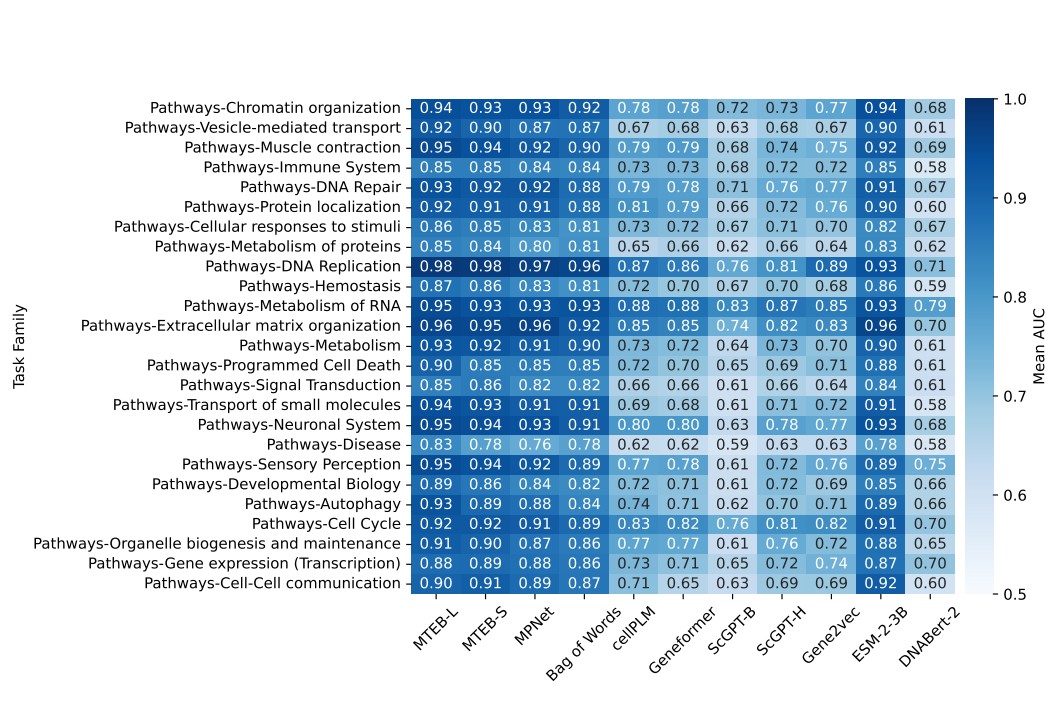

Figure S7: Model performance measured by mean AUC for binary tasks derived from the multi label task 'pathways'

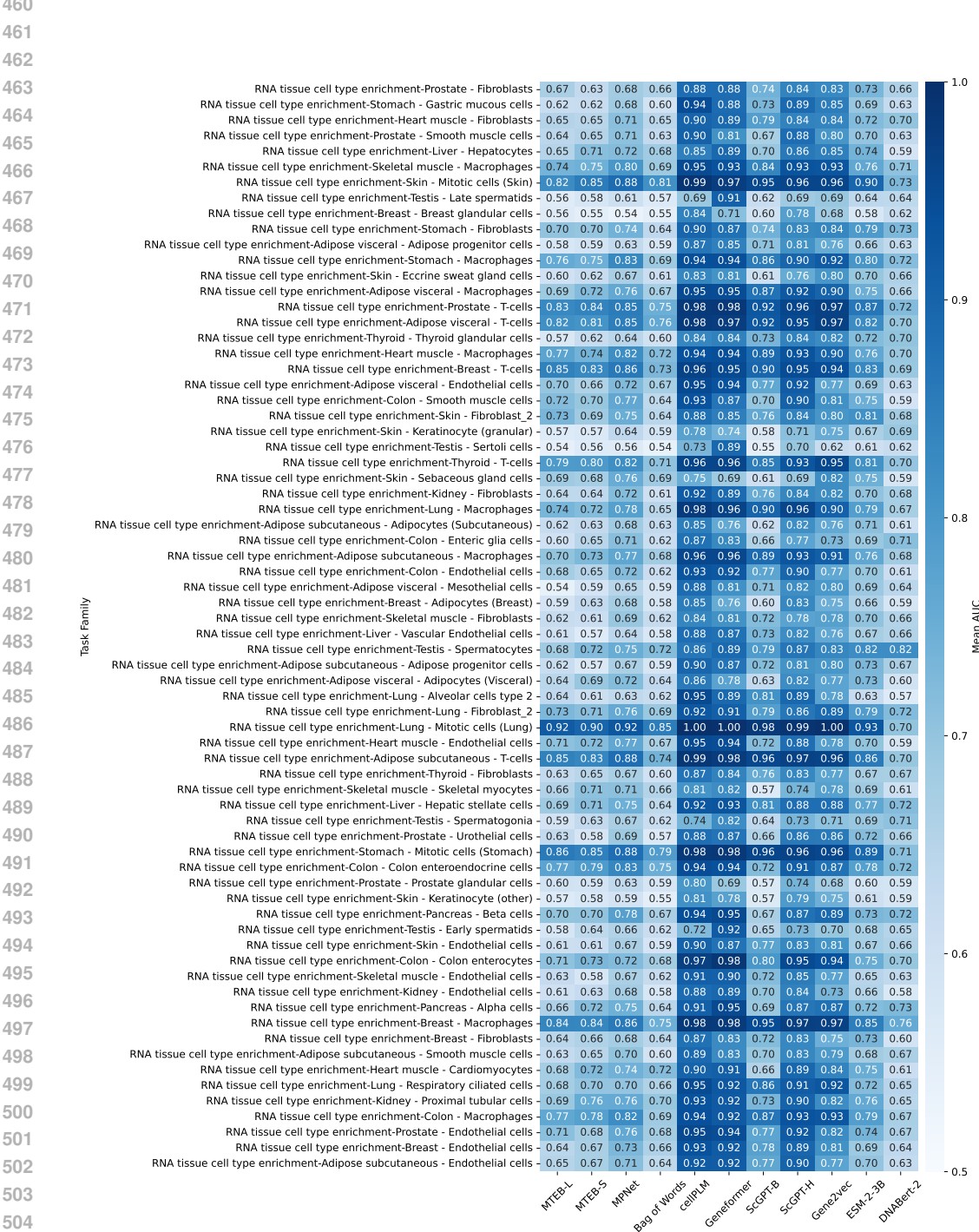

Figure S8: Model performance measured by mean AUC for binary tasks derived from the multi label task 'RNA tissue cell type enrichment'

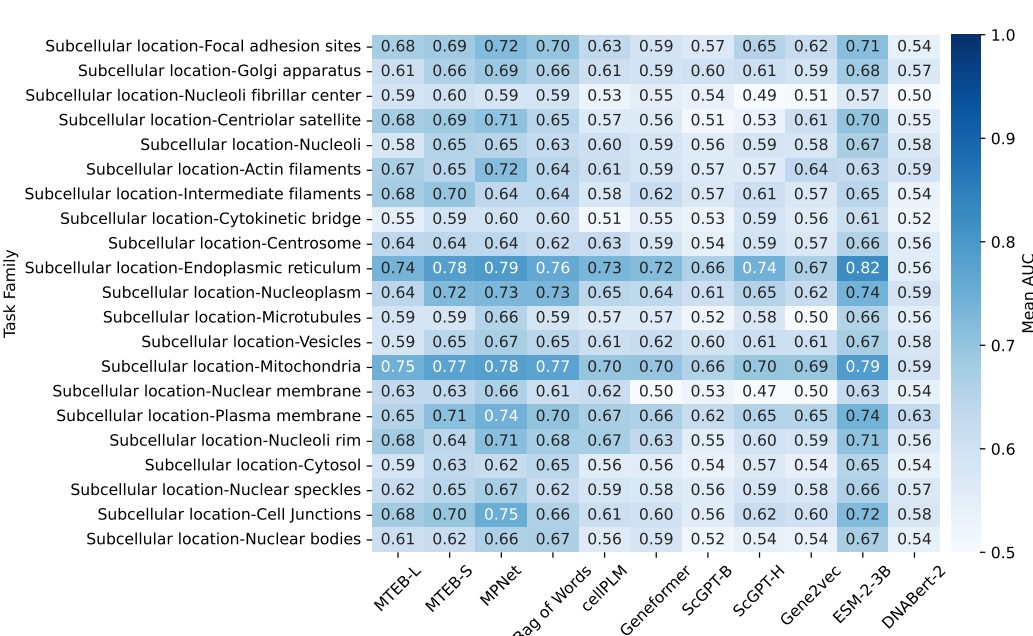

Figure S9: Model performance measured by mean AUC for binary tasks derived from the multi label task 'sub cellular location'

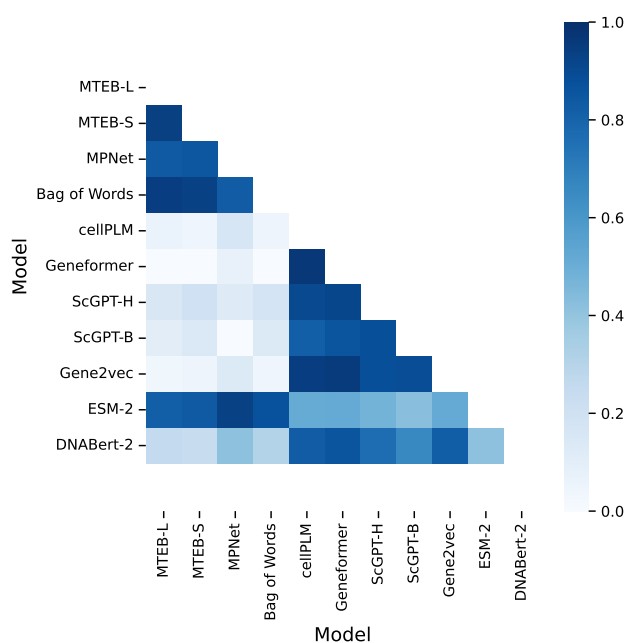

Figure S10: Similarity of performance across models. We construct vectors of the average AUC ROC for every model and task and then use 1 - cosine distance vectors to calculate their proximities, which are then re-scaled to the interval (0,1).

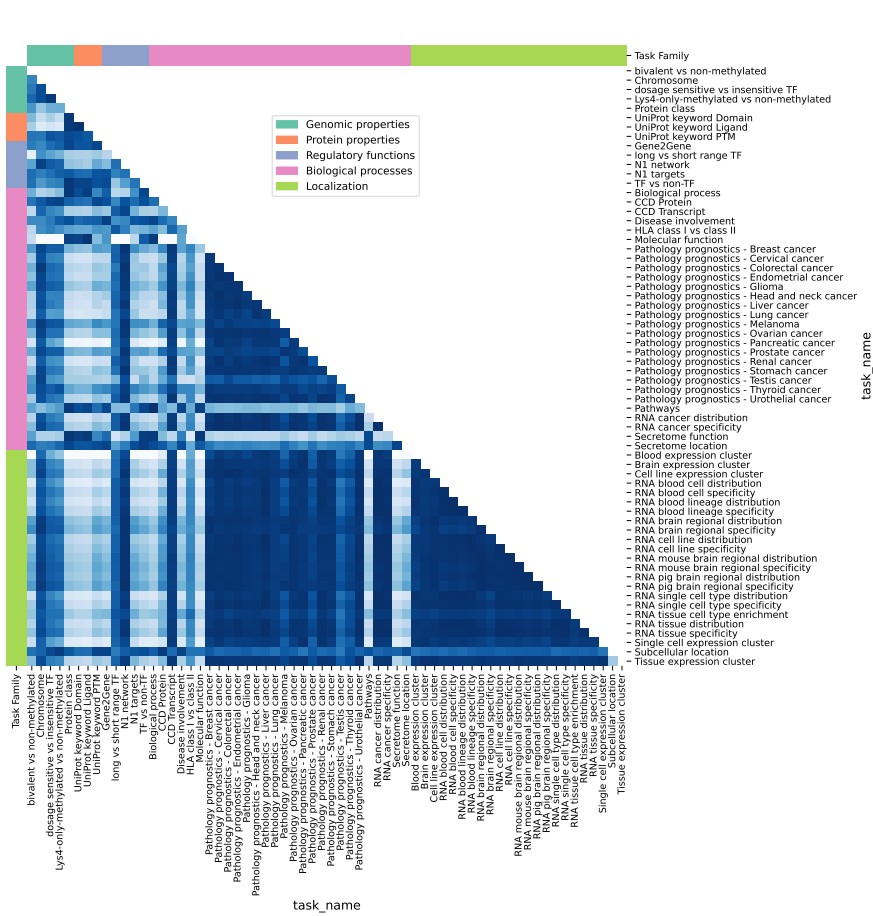

Figure S11: Similarity of performance across tasks. We construct vectors of the average AUC ROC for every model and task and then use 1 - cosine distance vectors to calculate their proximities, which are then re-scaled to the interval (0,1).

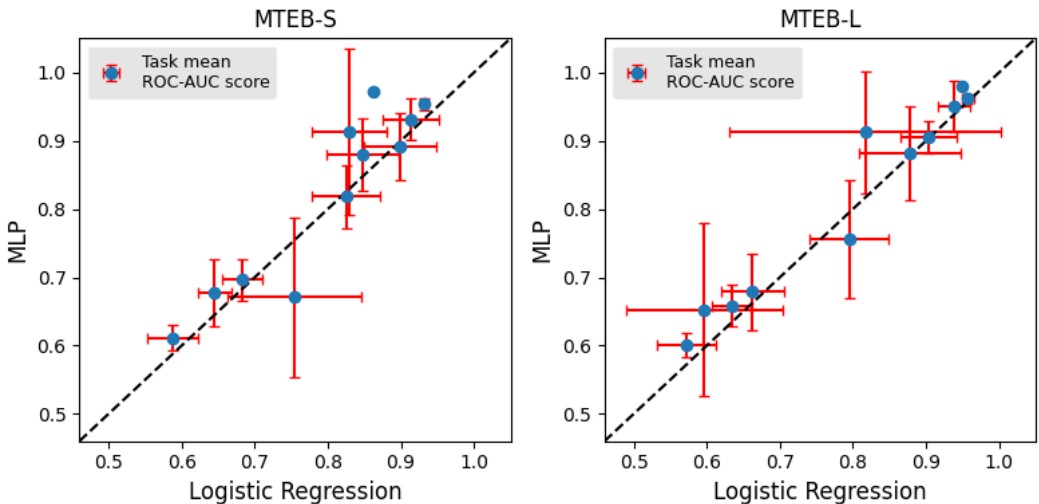

Figure S12: The prediction performance as measured by the 5-fold mean AUC-ROC score (with standard deviation in red) of a multiplayer perceptron (MLP) model versus a logistic regression model using the embeddings of MTEB-S and MTEB-L. We can see that in both cases the correlation is high (Pearson's coefficient of 0.92 and 0.97) and significant (p-value<0.001).

