# OpenReview forum: "Does your model understand genes? A benchmark of gene properties for biological and text models"
_ICLR.cc/2025/Conference — Submitted to ICLR 2025_

### Official Review · Reviewer_doRJ · 2024-10-24

**Soundness:** 2
**Presentation:** 3
**Contribution:** 2
**Rating:** 5
**Confidence:** 5

**Summary:**

This paper benchmarks several biological foundation (and text-based) models on a wide array of biological tasks. This is achieved by first extracting gene embeddings from these models and then training simple classifier/regression models to predict a target value. The benchmark carries importance in helping direct research efforts towards more effective models.

**Strengths:**

- The addition of text-based as well as models trained on base pairs and protein sequences is important in assessing the relative strengths of such tools against models trained on scRNA-seq data alone.
- The list of tasks selected for benchmarking is comprehensive.
- The paper is easy to read and understand. Objectives are stated clearly.

**Weaknesses:**

Below I provide some comments that if addressed, I think would strengthen this paper.

- Some models, such as Geneformer, are meant to be fine-tuned on downstream tasks, which can lead to a boost in accuracy. The current work relies only on pretrained embeddings which may limit its utility.
- Text-based models appear to perform well on many tasks related to regulatory functions (e.g., TF vs non-TF) and protein class families. The NCBI summaries used as input for these models often contain explicit information about TFs/targets/protein classes, which is what these models are being evaluated on. It may be misleading to claim these models are better suited for these tasks if the information is handed to them rather than learned. Would be interested to know how much explicit information is actually contained in the summaries.
- Many biological processes involve only a small number of genes (say, a few tens out of 20k genes), resulting in highly imbalanced distributions for the binary tasks under the “biological process” family. It would help to inspect these tasks closely and show some confusion matrices to detect any potential issues. Also, stratified cross-validation could prove crucial for interpreting F1 scores. It is unclear if this is what the authors implemented.

**Questions:**

- The number of genes used for the benchmark is not mentioned.
- Intuitively, assuming similar scaling, there should be some correlation between similarity metrics, such as cosine similarity between two gene embeddings, and linear regression predictions. Why rely entirely on linear/logistic regression for task evaluation?
- How much does the dimensionality of the embeddings impact the performance of the predictive models? I would expect the performance of linear models to deteriorate with increasing dimensionality.
- Abstract is too long.
- It is mentioned in line 392 that “ScGPT-H was the top performer in two different families of tasks.” This contradicts Figure 2 where it does not appear to be the top performer in any task.
- The benchmark includes tasks for other species such as mouse and pig. This could be problematic as some models are only trained on human data (e.g., Geneformer).

---

### Official Review · Reviewer_CWmS · 2024-10-31

**Soundness:** 3
**Presentation:** 3
**Contribution:** 3
**Rating:** 5
**Confidence:** 5

**Summary:**

The paper introduces a benchmark to evaluate deep learning models on gene-related tasks across five property types: genomic, regulatory, localization, biological processes, and protein. Using gene embeddings from diverse models (RNA-seq, DNA, protein sequences, and text), it finds that text and protein models excel in genomic and regulatory tasks, while expression-based models are better for localization.

**Strengths:**

The paper is well-organized and easy to understand.

The paper introduces an interesting topic of using different source of data to understand genes.

**Weaknesses:**

The advanced methods, such as LLMs and rna-seq based methods are not better than bag-of-words in the proposed tasks. Therefore, it’s hard to say that further development methods can have better results in gene function prediction.

For the downstream tasks, the proposed methods still need an MLP to adopt a new concept or task, which limits the evaluation of the new input genes. Can the framework predict a new gene correctly?

Personally, I believe that what we can discover from a given gene embedding is more valuable. For example, whether we can discover the new pathways or whether the gene has new functions. However, the authors just use the gene embeddings as the input of a linear model, which is disappointing to me. After all, there is no guarantee that a simple linear model can transfer new gene embedding to a new situation.

**Questions:**

see weakness

---

### Official Review · Reviewer_3CN4 · 2024-11-03

**Soundness:** 2
**Presentation:** 2
**Contribution:** 2
**Rating:** 3
**Confidence:** 5

**Summary:**

This paper proposes a benchmark to compare deep learning models in biology, focusing on gene properties from curated bioinformatics databases. It evaluates models across tasks related to genomic properties, regulatory functions, localization, biological processes, and protein properties.

**Strengths:**

The paper does not have any particular strengths.

**Weaknesses:**

1. The motivation of the paper, which is based on the premise that no benchmarks exist to compare foundation models across modalities or to compare text models against models trained directly on biological data, does not sufficiently support the novelty of the proposed benchmark.
2. The paper claims that the benchmark can be found on GitHub, but neither the anonymous git link nor any supplementary files are provided. A basic requirement for a benchmark is to provide the code for reproducing all methods, as well as the associated data. While the dataset may be large, at least a sample of the data should be provided in the supplementary materials.
3. Without the provided code or a usable toolkit, it is not possible to evaluate the validity of the benchmark. Furthermore, the paper does not discuss similar tools, such as OpenBioLink, PyKEEN 1.0, or Open Biomedical Network Benchmark, among others. Including these comparisons would help frame the contribution of the benchmark.
4. There should be a dedicated section in the main text that describes the data, performs data analysis, and provides statistical figures and charts. Additionally, there is insufficient visualization of the data to analyze the different tasks and to show the distribution of the data across these tasks.
5. The dataset used in this paper is sourced entirely from public databases and has only been reorganized for the tasks at hand, contributing little to the field in terms of novelty.
6. In the appendix, the dataset descriptions should include proper citations for the sources of the data, and links to the servers where the datasets can be accessed should be provided.

**Questions:**

N/A

---

### Official Review · Reviewer_Mdrs · 2024-11-04

**Soundness:** 4
**Presentation:** 4
**Contribution:** 4
**Rating:** 6
**Confidence:** 4

**Summary:**

The authors create a new benchmark that evaluates the representations of genes on five tasks. It does so by training a linear probe on top of the embeddings. The benchmark evaluates models with inputs ranging from text to expression data and finds text and expression data models do best overall. Text-based models do best in genomic properties and regulatory function families while expression-based models do better in localization and biological process tasks.

**Strengths:**

The originality of the work comes from the breadth of the models considered in the benchmark. Comparing text to expression to sequence based models is interesting and encapsulates the increasingly multi-model approach to genomic-ML. Moreover, the authors cover many tasks incorporating many commonly used datasets. The paper is clear and easy to follow. I believe that genomic-ML will only increase in popularity (it seems every week there is a new sequence based model) and having a benchmark like the ones proposed by the authors will help the community to focus their efforts.

**Weaknesses:**

The main weakness was the benchmark does not consider the similarity of the pre-training sets of these models and the test set. Is it not possible that models have already seen test data before and then this leads to higher performance?

It would also be interesting to test an ensemble model where all best models output results and then vote for the result and see how that helps. In the paper multi-modality was implied as a future direction, so this analysis could help to begin to move in that direction.

It would also be good if the authors evaluate EVO and other long-context protein language models as well as this is a quickly growing slice of this space of genomicML.

For some tasks it seems no model did well. Have the authors considered dataset size and other factors that could influence model performance? Perhaps these tasks do not have a lot of data to begin with.

**Questions:**

No questions.

---

### Meta-Review · Area_Chair_BbwY · 2024-12-19

**Metareview:**

This paper opens a new interesting avenue in benchmarking models in biology by combining LLM and bio foundation models. Taking data from curated databases a large number of classification tasks are defined. These are trained on representations coming from either LLM or bio foundation models.

The reviewers are mixed in their opinions leaning towards reject. Unfortunately, the authors did not engage in the discussion so the reviewers' comments are not answered. It is therefore hard to argue for acceptance in this case.

The benchmark and results raise important points well worth exploring so I hope this work will be updated and submitted elsewhere.

**Additional Comments On Reviewer Discussion:**

None.

---

### Decision · Program_Chairs · 2025-01-22

Reject